# Potential Effects of Garlic (*Allium sativum* L.) on the Performance, Immunity, Gut Health, Anti-Oxidant Status, Blood Parameters, and Intestinal Microbiota of Poultry: An Updated Comprehensive Review

**DOI:** 10.3390/ani14030498

**Published:** 2024-02-02

**Authors:** Wafaa A. Abd El-Ghany

**Affiliations:** Poultry Diseases Department, Faculty of Veterinary Medicine, Cairo University, Giza 12211, Egypt; wafaa.soliman@cu.edu.eg; Tel.: +20-01224407992

**Keywords:** *Allium sativum*, meat and egg production, antibodies, anti-oxidant enzymes, cholesterol

## Abstract

**Simple Summary:**

Using antibiotics as growth promoters or antimicrobials is a potential health hazard. Garlic (*Allium sativum* L.) has been extensively used in several aspects of poultry production systems. Therefore, this review article discusses the impact of using garlic as a feed additive on the performance, immunity, gut health, anti-oxidant status, blood parameters, and intestinal microbiota of poultry. Garlic feeding has been regarded as a potential antibiotic-alternative feed additive due to its great benefits to the health of poultry.

**Abstract:**

The use of antibiotics as growth promoters or for the prevention of some poultry diseases has faced global concern and serious criticism. Their addition to poultry feed has shown hazardous effects, including the development of antimicrobial resistance and a potentially harmful effect on human health. To eliminate these threats, there is increasing interest in natural alternatives. Plant derivatives such as garlic (*Allium sativum* L.) and its derivatives are presently extensively used in the poultry production system. The dietary supplementation of broilers and layers with garlic induced improvement in the production parameters, carcass quality, and intestinal integrity. The modulation of the immune response against some important viral diseases has resulted from the supplementation of poultry with garlic. Moreover, garlic has been shown to modulate gut health through antibacterial and antiparasitic activities. Treatment with garlic can also mitigate oxidative stress and reduce free-radical production. The reduction of cholesterol levels and improvement of some liver and blood parameters were also reported following the dietary inoculation of garlic. This review was designed to investigate the influence of garlic as a dietary additive on the performance, immunity, gut health, anti-oxidant status, blood parameters, and intestinal microbiota of poultry.

## 1. Introduction

As a result of the worldwide ban on antibiotic growth promoters, attention has turned toward finding alternatives without resistance or residues [1]. Through a global trend to go back to nature, the World Health Organization has encouraged the use of natural phytogenic substances to replace or reduce the use of antibiotic growth promoters. Phytobiotics, or phytogenics, are plant derivatives that have been used as feed additives to improve the health and performance of animals [2]. Over the past decade, this safe source of active ingredients has been regarded as an attractive research subject and has shown promising results [3]. Herbal plants possess multiple therapeutic properties and different effects.

Garlic (*Allium sativum*) is a perennial bulb-producing plant that belongs to the genus *Allium* in the family *Liliaceae*. Since antiquity, garlic has been grown on a large scale in all countries and has been widely used as a feed additive and growth promoter [4]. It has a specific smell and taste, as well as therapeutic properties in alternative medicine [5]. 

Garlic is estimated to contain different bioactive compounds, including organosulfur compounds diallyl thiosulfonate (allicin), diallyl sulfides, diallyl disulfide, diallyl trisulfide, and S-allyl-cysteine sulfoxide (alliin), saponins, phenols (β-resorcylic acid, pyrogallol, gallic acid, rutin, protocatechuic acid, and quercetin), amino acids, polysaccharides (fructose, glucose, and galactose), essential oils, vitamins (ascorbic acid, ribofavin, niacin, thiamine, and folic acid), minerals (germanium, selenium, phosphates, calcium, and iron), and enzymes [6,7,8,9]. The chemical structure of garlic is illustrated in Figure 1. The allin and alliinase enzymes collaborate to produce allicin [10], which is released from its precursor form when garlic bulbs are crushed or destroyed in digestion. Allicin, or daily thiosolphinic acid, is an active inhibitory principle of garlic [11]. Moreover, allicin ingredients can decompose, forming many volatile organosulfur compounds with bioactivities [12]. The nutritional value of raw garlic is represented in Table 1. However, little information exists on the effects of garlic products on nutrient utilization in poultry.

Garlic contains more than 200 chemical substances that are used for the prevention and treatment of cardiovascular disease [13], as well as anti-oxidants [14], antimicrobial [15,16], anti-inflammatory [17], anti-atherosclerotic, anti-thrombotic, anti-hypertensive, anti-diabetic, anti-cancer, and hypoglycemic properties [16,18,19]. The most important immune-modulating compounds in garlic are polysaccharides. The metabolism of fungi can be interrupted by garlic oil through the production of key genes represented in oxidative phosphorylation, cell cycle, and the processing of protein in the endoplasmic reticulum. Moreover, fungal growth could be hindered via the penetration of garlic oil into cells, causing the destruction and escape of cytoplasm and macromolecules [20].

Garlic could be given to poultry in the form of powder, aqueous extract, essential oil, and other commercial products either in the feed or in the drinking water. Dietary feeding of poultry on garlic resulted in enhancement in growth performance, gut health, dressing yield, and production cost [21,22,23], modulation of immunity and blood parameters [14], prevention of bacterial and parasitic infections [24,25], and mitigation of heat stress [26]. The addition of garlic to the broilers’ feed has no negative influence because it does not leave any residue, and the birds’ manure does not contaminate the environment. Therefore, products from garlic-consuming animals are safe for consumption.

The objectives of this review article were to investigate the findings on the influence of garlic as a dietary additive on the performance, immunity, gut health, anti-oxidant status, blood parameters, and intestinal microbiota of poultry.

## 2. The Different Influences of Dietary Garlic on Poultry Health

### 2.1. Production Parameters

#### 2.1.1. Performance 

The different effects of dietary garlic on the production performance parameters of broilers and layers are presented in Table 2 and Table 3 [27,28,29,30,31,32,33,34,35,36,37,38,39,40,41,42,43,44,45,46,47,48,49,50,51,52,53,54,55,56]. Inoculation of garlic in the diets of birds could enhance the production performance parameters, including feed intake (FI), body weight (BW), body weight gain (BWG), and feed conversion ratio (FCR) [57]. The mechanism by which the garlic powder can improve these parameters could be related to the presence of several organosulfur components, including allicin, alliin, ajoene, dithiin, diallyl sulfide, and S-allyl cysteine [58]. Similarly, the study of Ross et al. [59] demonstrated that the antibacterial compound dialkyl polysulfide in garlic plays a central role axial role in improving the BWG in broilers. A combined diet containing garlic and turmeric (10 g/kg each) reduced the pH of the digestive tract and enhanced apparent and digestible metabolized energy in the ileum of broiler chickens [60]. Moreover, garlic might increase the performance of pancreatic enzymes, which creates a good environment for nutrient digestion and absorption [14]. 

The FI of broilers [61] and layers [49] increased by increasing the level of garlic powder inoculation in the diet. This result may be owed to the high content of garlic to aromatic oil that enhances the digestion process. 

#### 2.1.2. Intestinal Architecture

The addition of eugenol and garlic tincture could improve intestinal integrity and enhance mucin-producing goblet cell numbers as a defensive response in birds against necrotic enteritis [62]. The inoculation of broilers diets with garlic at concentrations of 0.125, 0.25, 0.5, and 1% significantly increased the villus height and crypt depth and reduced the epithelial thickness and goblet cell numbers in the intestines of birds [63]. Moreover, the highest capacity of crypts and villi in small intestines was detected following the dietary addition of garlic in coccidiosis-infected broilers [64]. Allicin can regenerate and improve the physiological structure of the intestinal epithelium layer and increase the crypt’s depth and villus height, which eventually supports the digestive capacity by increasing nutrient absorption and assimilation. Elongated villi with deep crypts is considered an indication of a vigorous intestine architecture and, consequently, a good digestive capacity and pancreatic enzyme activity. Furthermore, the anti-oxidant characteristics of garlic can enhance overall gut function and improve nitrogen energy utilization [65]. Yang et al. [66] reported that the feeding of broilers on garlic reduces the pH of digesta, which increases the production of volatile fatty acids and the proliferation of beneficial bacteria. The dietary addition of 0.5% garlic efficiently reduced systemic hypertension and the prevalence of ascites but had no negative influences on broiler performance [36]. The inulin component of garlic decreases the digesta pH of birds and increases the volatile fatty acid production, which may help enhance beneficial bacterial colonization [67].

Others showed that garlic supplementation does not affect the feed efficiency or growth performance of broilers and layers [44,46,55,68]. This discrepancy might have resulted from the variances in the experiment duration, birds’ genetic and health conditions, and the form, treatment, and quality of garlic end-product components.

### 2.2. Immunity

The effect of garlic on the immunity of birds is illustrated in Table 4. It has been found that the different forms of dietary garlic alone or in combination with other aromatic phytobiotics can enhance the immune response in terms of enhancing antibody titers against and increasing the immune organ:body weight ratio [14,26,38].

The improvement in the immune response may be related to the characteristics of biologically active compounds in essential oils, such as antimicrobial, anti-oxidant, and anti-inflammatory properties, which provide essential nutrients for the development of the immune cells. In addition, promoting the proliferation of lymphocytes in the primary immune organs and improving intestinal integrity could stimulate the production of immunoglobulin (Ig), such as IgG, IgM, and IgA, which is associated with increasing the relative weight of the immune organs. Many immuno-stimulator compounds are present in garlic, including the lectin family, which is known to interact with pathogen recognition receptors on immune cell surfaces [75]. Garlic is one of the impressive conductors of the body’s immune system, which stimulates the immune function by making macrophages or killer cells more active. Moreover, garlic can improve humoral immune cell functions via the enhancement of cytokine production and/or antigen-presenting cell phagocytic capacity [70]. Dorhoi et al. [76] demonstrated that a high dose of garlic extract (200 mg/mL) on a macrophage culture of laying chickens could impair the phagocyte function and inhibit phagocytosis, whereas a low dose (50 mg/mL) increased sheep red blood cells count. Inoculation of garlic extract or its protein fraction increased the destruction in peritoneal macrophages and the engulfment of parasites in Leishmanial major-infected Balb [77].

Polysaccharides of garlic show an immune potentiation mechanism through the regulation of interleukin (IL)-6, IL-10, tumor necrotizing factor-α, and interferon-gamma (INF-γ) expression in RAW 264.7 macrophages. In addition, garlic extract could augment concanavalin A (ConA)-induced splenocytes, thymocyte proliferation, and the gene expression of IL-2 and INF-γ in vitro [78]. Moreover, the addition of garlic extract to a culture augmented the induction of IL-2 and IL-12, INF-γ, and tumor necrosis factor α in stimulated splenocytes [79]. Low concentrations of diallyl trisulfide (3–12.5 mg/mL) of garlic enhanced the proliferative reactions in a culture, while higher concentrations (50 mg/mL) inhibited T-lymphocyte proliferation in mice [80]. Aged garlic extract stimulated the proliferation and increased the activity of T-cells and natural killer cells, as well as enhancing phagocytosis and cytokine release [81,82].

Garlic supplementation increased the relative weights of immune organs, such as the spleen, thymus, and bursa of Fabricius, the white blood cell count, as well as lymphocytes, splenocytes, and thymocyte proliferation [70]. In addition, the titers of antibodies against Newcastle disease virus (NDV), sheep red blood cell count, and *Brucella abortus* (BA) have been increased following the administration of garlic in laying chickens [70].

It has been demonstrated that the anti-oxidative stress of garlic is a potential factor that enhances the immune response [83]. Supplementation with garlic extract at 4 and 8 mg/mL revealed that macrophages may display antimicrobial activity and enhance the production of reactive oxygen species.

### 2.3. Gut Health

#### 2.3.1. Antibacterial

Garlic oil is regarded as a major antibacterial component that disturbs both the structure and metabolism of bacterial cells. The strong antimicrobial effects of garlic have been reported [40,58,84,85]. It has been reported that garlic extracts exert a differential inhibition between beneficial intestinal microflora and potentially harmful enterobacteria [86]. Garlic could reduce the number of gut-pathogenic bacterial populations such as *Escherichia coli* (*E. coli*). Garlic showed an ability to inhibit *E. coli* 10 times greater than that seen in *Lactobacillus casei* [87]. Rahimi et al. [71] demonstrated that *E. coli* count was significantly reduced in the digesta of ileo-cecum of broiler chickens supplemented by a blend of garlic, thyme, and coneflower. Recently, Elbaz et al. [23] found that garlic treatment could reduce the ileal enumeration of *E. coli* and total coliform but increase the *Lactobacillus* count. In addition, the positive influences of eugenol and garlic mixture on broiler performance and intestinal health status under necrotic enteritis conditions have been reported [85]. Microencapsulated eugenol and garlic tincture modulated the microbiota balance by inhibiting pathogenic growth while promoting beneficial microbial growth, as well as reducing the severity of the intestinal lesions of broilers with necrotic enteritis [62]. The antimicrobial and bacteriostatic properties of garlic extract are associated with the presence of an allicin-active compound [15]. Allicin exhibited a bacteriostatic effect on some vancomycin-resistant enterococci. In addition, allicin displays SH group reactivity on cysteine residues, causing deactivation and suppression of specific thiol-containing enzymes in pathogens [88]. This reaction induced deactivation and suppression of specific thiol-containing enzymes in pathogens [24,89]. Garlic is a nucleophilic agent that can counteract the impact of electrophilic substances on micro-organisms [90].

#### 2.3.2. Antiparasitic

The in vitro and in vivo anticoccidial activities of different processed extract forms of garlic have been documented [91,92,93,94,95,96,97,98,99]. The study of Ali et al. [100] found that supplementing coccidiosis-infected broiler chickens with garlic at 15 g/kg feed reduced the oocyst shedding and lesion score but improved the histopathology of the small intestines. In the same context, continuous feeding of *Eimeria tenella*-infected broiler chickens on natural garlic essential oil (0.06 mL/L drinking water) significantly reduced the clinical signs, cecal lesion score, and the oocyst shedding but increased the weight of diseased chickens and effectively improved the intestinal functions [74]. In comparison with ginger oil, garlic oil (150 μL/100 mL) showed superior efficacy against the *Eimeria* species infection of quails in terms of improved activity level with better health, increased feed intake, and complete recovery from oocysts on day 15 post-infection [101].

Allen et al. [102] reported that the anti-oxidant properties of garlic cause oxidative stress against parasites and neutralize oxygen-reactive species. Furthermore, Pourali et al. [103] have attributed the anticoccidial activity of garlic to its immunomodulatory activity. Similarly, Kim et al. [104] revealed enhanced protection from *Eimeria acervulina* infection in chickens after feeding on garlic metabolites [104]. Propyl thiosulfinate oxide and propyl thiosulfinate active ingredients of garlic reduced fecal oocyst shedding and enhanced the antibody response against coccidial infection [104]. Likewise, the aqueous garlic extract is rich in phenols, flavonoids, and varying sulfur compounds [16]. The phenolic compounds change the permeability of the cytoplasmic membrane to many cations, inhibit the physiological functions, and, consequently, result in the loss of membrane potential, allowing vital cellular substances to leak out, protein and ATP production to be inhibited, and cellular death to occur [105].

Allicin induces changes in the intestinal microbiota, exerts an anti-oxidant effect on *Eimeria* oocysts, and stimulates immunity by enhancing the antibody response, which directly destructs sporozoites [57,106]. Additionally, the phenolic component in garlic acts on the cytoplasmic membrane of *Eimeria* species and makes changes in their cation permeability, leading to the death of *Eimeria* [107]. Moreover, allicin interrelates with the cytoplasmic membranes of the intestine, changes the permeability of cations, disturbs the internal vital processes of cells, and, finally, causes the death of the parasite [108]. The capability of allicin and alcoholic garlic extract to inactivate the oocysts of *Eimeria tenella* makes them preferable to chemical disinfectants [25]. *Eimeria* oocysts sporulated in allicin-containing media exhibited the lowest post mortem lesion score and oocyst count shedding when compared with oocysts sporulated in alcoholic garlic extract and potassium hydroxide [25]. Doses of 360 mg/mL garlic extracts and 180 mg/mL allicin significantly reduced oocyst numbers by 73.5 and 88.3%, respectively [25].

Moreover, garlic crude extract showed great activity against worms and protozoon parasites *Cryptosporidium* spp. in different animal models [109,110,111].

### 2.4. Anti-Oxidant Status

Garlic exhibited strong anti-oxidant activity in birds (Table 5) [112,113,114,115,116,117,118,119]. Phenols and saponins, which are components of garlic, have strong anti-oxidant effects. For instance, saponins could inhibit the growth and DNA destruction induced by H_2_O_2_. Consequently, protected mouse-derived myoblasts were able to scavenge intracellular reactive oxygen species [120]. The imbalance between the oxidation and reduction in the host’s cells induces significant destruction of them with subsequent oxidative stress. However, the anti-oxidant enzymes can prevent the free radicals from attacking cell membranes [121]. Essential oils, present in different aromatic plants, contain several natural anti-oxidants [14]. Garlic and/or garlic tocopherol induced a much higher anti-oxidant effect by reducing the production of free radicals [14,46], especially in birds under heat-stress conditions [26].

Decreased actions of hydroxymethylglutaryl coenzyme A reductase, cholesterol 7 α-hydroxylase, and fatty acid synthetase have been demonstrated after the administration of garlic powder polar fractions (garlic equivalent to 1, 2, 4, 6, and 8% fresh garlic paste) [122]. The diallyl polysulfides from an aged garlic extract could protect the cell membranes from lipid peroxidation [123]. Moreover, essential oils present in garlic and other plants can remove oxygen free radicals by reducing the level of malondialdehyde (MDA) and enhancing the levels of superoxide dismutase (SOD) and glutathione peroxidase (GPx) [103,124,125].

### 2.5. Blood Parameters

The influence of the dietary addition of garlic on the different blood parameters of poultry is shown in Table 5.

Many studies showed the hypocholesteric effect of garlic in broilers and layers [14,26,54,56,117]. Garlic-containing enzymes may have a role in regulating the metabolism of lipids and enhancing enzyme activities that stimulate biliary cholesterol secretion and lower the fractional absorption of dietary cholesterol [22]. Moreover, the inhibition of acetyl CoA synthetase and 3-hydroxyl-3-methylglutaryl-CoA reductase enzymes, which are required for cholesterogenesis and the biosynthesis of fatty acids, can reduce the blood cholesterol level [126]. Similarly, the potential effect of garlic on the lipid metabolism in layers may be related to the reduction of lipogenic and cholesterogenic-depressing effects of some hepatic enzymes, such as fatty acid synthase, glucose 6 phosphatase dehydrogenase, and malic enzyme, and consequently, the mechanism of hypocholesterol and hypolipid syntheses [53]. Lower serum and liver cholesterol [122] inhibits bacterial growth [127], reduces platelet formation, and decreases oxidative stress [123].

Garlic oil could improve the anti-oxidant enzyme activities in the liver, inhibit 1,3-dichloro-2-propanol metabolic activation, and reduce hepatic apoptosis, thus protecting against liver damage [128]. In addition, organosulfur compounds in garlic could treat liver damage by decreasing the release of hepatic pro-inflammatory cytokines and enhancing anti-oxidant activity by suppressing cytochrome P450 2E1 expression [129,130].

Additionally, the effect of garlic on hematological parameters such as red blood cell (RBC) and white blood cell (WBC) counts, and hemoglobin and packed cell volume have been reported [14,34,35]. The hemolytic bioactives and their metabolites in garlic can be the main causes of these effects. Increasing erythrocyte count with garlic supplementation could be due to the synthesis of RBCs following the formation and secretion of renal erythropoietin [131]. Moreover, the addition of garlic extract to the diets of laying hens could improve the uptake of splenic RBCs [76].

### 2.6. Intestinal Health and Microbiota

Garlic powder, garlic meal, and garlic derivatives have improved intestinal health status, which may contribute to the improved intestinal morphology of treated broilers [132,133]. The addition of garlic powder successfully reversed the damaged intestinal morphology of lipopolysaccharide-challenged broilers in terms of increased villus height, intestinal health, and growth efficiency [134].

The intestinal microbiota in broiler chickens plays a key role in the health and growth of birds [135]. There is little and contentious information regarding the effects of garlic derivatives on broiler intestinal microbiota. Nevertheless, garlic and garlic products have been found to be effective against several pathogenic bacteria causing enteritis [136]. The garlic derivative propyl-propane thiosulfonate showed antimicrobial activity against enterobacteria, *E. coli*, *Salmonella* spp., and *Campylobacter jejuni*. It has been shown that propyl-propane thiosulfonate can modulate the intestinal microbiota composition and improve the nutrient digestibility of growing broilers [137]. Moreover, a significant reduction in *Clostridium coccoides*/*Eubacterium rectale*, and *Clostridium leptum* log_10_ number of copies, while increasing in bacteroides and total bacterial contents, were observed in ileum following feeding on 11.3% propyl-propane thiosulfonate [137].

Exposure of broiler chickens to 0.5% *A. hookeri* leaf resulted in differences in the abundance of gut microbiota genera compared to diets containing 0.3% [138]. A diet containing 0.5% *A. hookeri* leaf reduced the profusion of *Eubacterium nodatum*, *Marvinbryantia*, *Oscillospira*, and *Gelria* [138]. This effect may be related to the abundance of pharmacologically active components in garlic, such as organosulfur, polyphenols, and allicin, that are known to affect the gut microbiota by enhancing or suppressing bacterial configuration [139]. A high percentage (~90%) of absorbable polyphenols are digested in the intestine by microbiota rather than the digestible enzymes [140]. Moreover, allicin is an organosulfur compound used against various bacterial pathogens, including *Staphylococcus* and *Pseudomonas* [141]. The antibacterial activity of allicin is related to the chemical interaction with thiol groups in enzymes. These enzymes are important for the metabolic activities of cysteine proteinase, which influences bacterial virulence and the antibacterial effect [142].

## 3. Conclusions

The supplementation of garlic to broiler and layer poultry species mostly shows improvement in performance and production efficiency, enhancing the immune response, maintaining gut health, reducing exudative stress, and modulating many important blood parameters. However, the different modes of action of garlic are indefinite. Therefore, further studies should focus on establishing the mechanisms of actions of garlic and its derivatives.

## Figures and Tables

**Figure 1 animals-14-00498-f001:**
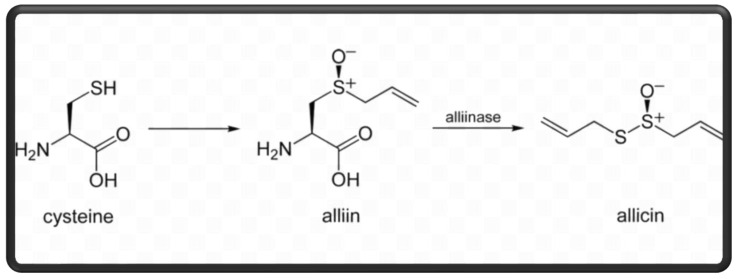
Chemical structure of garlic.

**Table 1 animals-14-00498-t001:** The nutritional value of raw garlic/100 g.

Component	Importance
Carbohydrates (33.06 g)	Important for energy, immunity, disease prevention, and blood clotting
Protein (6.36 g)	Development of body tissues
Fiber (2.1 g)	Shortens the stagnant time in the gut
Sugar (1 g)	Important for energy, immunity, disease prevention, and blood clotting
Fats (0.5 g)	Formation of cell membranes
Vitamin B3 (0.7 g)	Formation of coenzyme NAD
Vitamin B5 (0.6 g)	Formation of coenzymes of amino acid metabolism
Vitamin B2 (0.1 g)	Formation of coenzyme FAD
Vitamin B1 (0.2 mg)	Carbohydrate metabolism and synthesis of acetylcholine
Vitamin B6 (1.2 mg)	Formation of coenzymes in different reactions
Vitamin C (31.2 mg)	Protein synthesis
Vitamin B9 (3 µg)	Synthesis of DNA
Calcium (181 mg)	Formation of bone and coagulation process
Phosphorus (153 mg)	Formation of lipids, proteins, sugars, and nucleic acid
Magnesium (25 mg)	Cofactor for kinase and decarboxylase
Sodium (17 mg)	Formation of membrane
Zinc (1.16 mg)	Cofactor for some enzymes
Selenium (14.2 µg)	Cofactor for glutathione coagulase
Sulfur (16%)	Antimicrobial

**Table 2 animals-14-00498-t002:** The different effects of dietary garlic on the production performance parameters of broilers.

Dose/Route	Effects	Reference
Garlic paste (3.8%), solvent fractions, or garlic oil equal to this quantity in feed	No effect on FI	[27]
Garlic 0, 0.01, 0.1 or 1% in meal	No improvement in the performance	[28]
Garlic powder 0.2% and 0.4% of feed	No effects on BWG, FI, FCR, carcass cuts, and visceral organs	[29]
Garlic at 1 kg/ton feed	Enhanced carcass yield	[30]
Garlic 1, 3, and 5% and 3% garlic powder + 200 IU of α-tocopherol/kg of feed	No influence on performance Increased crude protein Decreased crude fat contents of carcass, the pH, and thiobarbituric acid reactive substances of meat	[31]
Garlic 0.5%, 1.0%, and 3%	Decreased heart weight	[32]
Garlic powder 0.5% of feed	Increased live BWG	[33]
Garlic powder 3% and 5% of diet	Increased breast weight (3%) Low BW (5%)	[34]
A mixture of ginger and garlic (1:1 ratio) 50 mL/L of the drinking water	Improved BW, BWG, FI, and FCR	[35]
Garlic bulb 5, 10, or 15 g/kg meal	Decreased BW (high dose and standard temperature) No effect on the FCR	[36]
The 5 g/kg garlic powder + 1 g/kg black pepper powder and 10 g/kg garlic powder + 2 g/kg black pepper powder	Improved WG and FCR	[37]
Fresh garlic paste 0.2, 0.4, 0.6, and 0.8%/L of drinking water	No effect on BWG or FCR Decreased mortality	[38]
Garlic powder 3% in diet and a mixture of garlic powder 1.5% plus turmeric powder 0.25%	Improved BWG, FI, FCR, performance index, and protein efficiency ratio	[39]
Garlic paste 0.25% and 0.50% with basal diet	Improved BWG, FCR, and livability No influence on carcass attributes	[40]
Garlic 5 g/kg feed, black cumin 5 g/kg, or their combination	No difference in BWG, FI, FCR, and relative organ weights	[41]
A basal diet plus 0.25, 0.50, and 0.75 g garlic powder/kg diet	Increased BW and BWG at 21 and 42 days of age High length and average width of small intestine	[14]
Garlic essential oil (200 mg/kg diet) alone/or in combination with lemon essential oil (200 mg/kg diet) under heat stress	Enhancement in BW, FCR, carcass dressing, and increasing the digestive enzymes Decreasing mortality rate and abdominal fat content	[26]
Garlic powder 3% of feed	Improved BWG and final BW	[42]

**Table 3 animals-14-00498-t003:** The different effects of dietary garlic on the production performance parameters of layers.

Dose/Route	Effects	Reference
1 or 3% garlic meal	Decreased egg yolk cholesterol	[43]
Garlic paste (3.8%), solvent fractions, or garlic oil equal to this quantity in feed	No effect on daily FI	[27]
Garlic oil 0.02% in meal	No effect on egg production, egg mass, body weight, feed consumption, and feed efficiency	[44]
Garlic powder 3% in diet	No differences in the color and flavor of eggs No change in yolk cholesterol concentrations	[45]
Sun-dried garlic paste 0, 2, 4, 6, 8, or 10% of diet	No effect on egg weight, egg mass, feed consumption, and feed efficiency among diets or birds’ strain Increased Yolk weight with increasing levels of dietary garlic Decreased yolk cholesterol concentrations	[46]
Garlic powder 0, 5, 10, and 15 g/kg feed	Decreased yolk weight	[47]
Garlic powder 0.5 and 10 g/kg feed	Increased egg weight Decreased egg yolk cholesterol triglyceride No effect on performance or egg albumin index, eggshell index, and egg Haugh unit	[48]
Garlic powder 0, 2, 6, or 8% in feed	Increased egg production	[49]
Garlic 2% and fenugreek 2%	No effect on FI, FCR, BW, BWG, egg rate, egg weight, and egg mass Increased yolk weight and color and Haugh units Decreased albumen weight	[50]
Garlic powder 8% in feed	Better egg production No effect on egg mass and egg weight	[51]
Garlic powder 1, 2, and 4% in feed	Increased egg production No effect on egg weight, yolk index, shell weight, shell thickness, yolk weight (1% garlic) Decreased eggshell index and Haugh unit (4% garlic)	[52]
Garlic juice at 0.25, 0.50, and 1%	Improved egg albumin, yolk and shell weight, albumin height, and Haugh unit	[53]
Garlic powder 1%, fenugreek 1%, and garlic powder 1% + fenugreek 0.5%	No effect on laying hens’ performance	[54]
Garlic 1, 2, and 3% of ration	No effect on BWG, FCR, egg production, egg mass, albumen weight, albumen height, Haugh unit, yolk index, yolk height, egg weight, fertility, hatchability, embryonic mortality, chick weight, and chick visual score, shell thickness, and shell weight An improvement in yolk diameter, yolk weight, chick length, and yolk color	[55]
A mixture of lemon, onion, and garlic juice in portions 1.00, 1.00, and 0.125/L of the drinking water, respectively	Improved FCR Increased number of eggs/hen, percentage of egg production, and egg mass/hen Enhanced yolk color and yolk percentage	[56]

**Table 4 animals-14-00498-t004:** The effect of dietary garlic on the immune response of poultry.

Dose/Route	Type of Production	Effects	Reference
Garlic powder 1% or 3% garlic	Broiler chickens	Enhanced antibody production against NDV and leukocyte count	[69]
Garlic 10 and 30 g/kg diet	White Leghorn chickens	Enhanced antibodies against NDV, SRBCs, and BA Augmented splenocyte and thymocyte proliferations Reduced CD4^+^, but increasing CD4: CD8^-^ lymphocyte ratios and WBCs count Increased relative weights of immune organs (spleen, thymus glands, and bursa of Fabricius)	[70]
Garlic 0.5%, 1.0%, and 3%	Broiler chickens	Lower weights of bursa of Fabricius and spleen	[32]
Garlic powder 0.1%	Broiler chickens	Improved relative weight of bursa of Fabricius without effect on the spleen weight No effect on NDV vaccine (LaSota) antibody response	[71]
Garlic powder 3% and 5% of diet	Broiler chickens	No influence on bursa of Fabricius and thymus weights Decrease spleen weight	[34]
A mixture of ginger and garlic (1:1 ratio) 50 mL/L of the drinking water	Marshal broiler chickens	Increased total protein, albumin, and globulin	[35]
Garlic extract (allicin) 25, 50, 75, or 100 mg/kg diet	Broiler chickens	Increased total protein and albumin concentrations by about 4.7 and 5.9%, respectively (50 mg/kg) No effect on total protein, albumin, or globulin concentrations (25, 75, or 100 mg/kg)	[72]
Fresh garlic paste 0.2, 0.4, 0.6, and 0.8%/L of drinking water	Broiler chickens	Increased antibody titer against NDV	[38]
Garlic meal 0.125% of feed	Broiler chickens	Reducing scores of IBDV signs Higher mortality rate High antibody response to IBDV	[73]
Garlic essential oil 0.06 mL/L drinking water	Broiler chickens	Improved immune organ index, IgM, IgG, and IgA	[74]
A basal diet plus 0.25, 0.50, and 0.75 g garlic powder/kg diet	Broiler chickens	Increasing total protein, globulin, IgM, and IgG Improved liver and immune-related organ weight	[14]
Garlic essential oil (200 mg/kg diet) alone/or in combination with lemon essential oil (200 mg/kg diet) under heat stress	Broiler chickens	Increasing the relative weight of bursa of Fabricius and the serum antibody titer against NDV No changes in relative weights of spleen and thymus glands, and antibody titer against AIV	[26]

**Table 5 animals-14-00498-t005:** The effect of dietary garlic on the anti-oxidant status and blood parameters of poultry.

Dose/Route	Type of Production	Effects	Reference
Garlic paste (3.8%), solvent fractions, or garlic oil equal to this quantity in feed	Broiler chickens Leghorn laying pullets	Decreasing serum cholesterol by 18 and 23% in broilers and Leghorn pullets, respectively	[27]
Garlic oil 0.02% in meal	Babcock B-300 strain of laying hens	No effect on serum cholesterol	[44]
Garlic 2% in feed	Broiler chickens	Lowering in hepatic cholesterol concentrations	[112]
Garlic 3% in meal	Broiler chickens	Decreased cholesterol in plasma and breast and thigh muscles	[113]
Garlic powder 3% in diet	Laying hens	No change in serum cholesterol concentrations	[54]
Sun-dried garlic paste 0, 2, 4, 6, 8, or 10% of diet	Hisex Brown, Isa Brown, Lohmann, Starcross, Babcock, and Starcross-579 strains of laying hens	Decreased serum cholesterol concentrations	[46]
Garlic 0, 1, 3, or 5% in meal	Laying hens	No change in HDL level	[114]
Garlic powder 0.5 and 10 g/kg feed	Laying hens	Decreased serum triglyceride	[48]
Garlic 2% and fenugreek 2%	Lohmann Brown laying hens	Increased HDL Reduced serum cholesterol and LDL	[50]
Garlic powder 1% or 3% garlic	Broiler chickens	No effect on leukocyte count	[69]
Garlic powder 10 and 20 g kg^−1^	Laying hens	Reduced total cholesterol, triglyceride, LDL, and HDL	[115]
Garlic powder 5–20 g kg^−1^	Broiler chickens	Decreased plasma LDL cholesterol No effect on HDL cholesterol	[116]
Fermented garlic powder 3% in diet	Laying hens	Decreased serum cholesterol	[117]
Garlic powder 1, 2, and 4% in feed	Laying hens	Increased plasma HDL and LDL (1, 2, and 4%).	[52]
Garlic 1, 3, and 5% and 3% garlic powder + 200 IU of α-tocopherol/kg of feed	Broiler chickens	Reduced the total and LDL levels Increased HDL levels	[31]
A mixture of garlic and thyme powder 0.1 and 0.2 g kg^−1^	Laying hens	No effect on cholesterol, triglyceride, HDL, and LDL	[118]
Garlic powder 0.1%	Broiler chickens	Decreased triglycerides, total cholesterol, and LDL Increased HDL	[71]
Garlic powder at 0.2% and 0.4% of feed	Cobb broiler chickens	Reduced triglycerides, cholesterol, and LDL Increased HDL	[119]
Garlic powder 3% and 5% of diet	Broiler chickens	Decrease spleen weight, RBCs, WBCs, and packed cells volume	[34]
Garlic powder 1%, fenugreek 1%, and garlic powder 1% + fenugreek 0.5% Garlic and fenugreek 2%	Laying hens	Decreased LDL Beneficial effects on cholesterol metabolism	[54]
A mixture of ginger and garlic (1:1 ratio) 50 mL/L of the drinking water	Marshal broiler chickens	Increased hemoglobin, packed cell volume, WBCs, RBCs, total protein, albumin, and globulin Decreased cholesterol	[35]
A mixture of lemon, onion, and garlic juice in portions of 1.00, 1.00, and 0.125/liter of drinking water, respectively	Bovan Brown layer chickens	Decreasing total plasma cholesterol content, GPT, GOT, and creatinine	[56]
Garlic 5 g/kg feed, black cumin 5 g/kg, or their combination	Ross-308 broiler chickens	Increasing total protein Reduced GOT	[41]
Probiotic, citric acid, and garlic supplemented with 0.5 g kg^−1^ multi-strain probiotic mixture, citric acid, and garlic powder, respectively. Probiotic-citric and probiotic-garlic groups treated with 0.5 g kg^−1^ multi-strain probiotic mixture and 0.5 g kg^−1^ citric acid and garlic powder, respectively, while citric-garlic group fed diet with 0.5 g kg^−1^ of citric acid and garlic powder.	Broiler chickens	Decreased cholesterol, triglycerides, and LDL Elevated HDL	[23]
A basal diet plus 0.25, 0.50, and 0.75 g garlic powder/kg diet	Broiler chickens	Increasing RBCs, hemoglobin HDL, SOD, and total anti-oxidant capacity Decreasing total cholesterol, LDL, GOT, and AMD	[14]
Garlic essential oil (200 mg/kg diet) alone/or in combination with lemon essential oil (200 mg/kg diet) under heat stress	Broiler chickens	Reducing MDA, triglycerides, cholesterol, and LDL Increasing HDL, SOD, and GPx	[26]

## Data Availability

All prepared data are presented in the present article.

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
