# Peer review of "Potential Effects of Garlic (Allium sativum L.) on the Performance, Immunity, Gut Health, Anti-Oxidant Status, Blood Parameters, and Intestinal Microbiota of Poultry: An Updated Comprehensive Review"

_animals, 2024, doi:10.3390/ani14030498_

Round 1
Reviewer 1 Report
Comments and Suggestions for Authors
Nice job writing this comprehensive review. Please consider the following:
1. In table 3, it states that garlic powder at 1 and 3% has no effects on antibodies for NDV and then it states that it enhanced antobodies for NDV. Please revise.
2. Please add a table and inscussion about the effects of garlic on intestinal health and intestinal microbiota
3. Please add a table containing the nutritional value and other metabolites contained in garlic. Also, include the most important varieties of garlic used for poultry.
3. Regarding the English language. Please make sure to go through the manuscript and correct the language as needed. Examples lines 18-19; 20-23 need to be reworded.
Comments on the Quality of English LanguageThe writing is, for the most part, okay but it requires small corrections to make sure the sentences make sense.
Author Response
Dear respected reviewer
Thanks for your valuable comments that will add a great value and help to improve the article.
- In table 3, it states that garlic powder at 1 and 3% has no effects on antibodies for NDV and then it states that it enhanced antobodies for NDV. Please revise.
- Corrected.
- Please add a table and inscussion about the effects of garlic on intestinal health and intestinal microbiota.
- There is a little information regarding the effects of garlic derivatives on the intestinal microbiota. However, the different effects of garlic on the intestinal health and microbiota have been added as a new section of the article (section 2.6).
- Please add a table containing the nutritional value and other metabolites contained in garlic. Also, include the most important varieties of garlic used for poultry.
- Added (Table 1).
- Regarding the English language. Please make sure to go through the manuscript and correct the language as needed. Examples lines 18-19; 20-23 need to be reworded.
- Corrected.
Best regards
Reviewer 2 Report
Comments and Suggestions for Authors
Review entitles Potential effects of garlic (Allium sativum L.) On the performance, immunity, gut health, antioxidant status, and blood parameters of poultry: An updated comprehensive review, describes the effects of garlic acid in poultry. Review paper is well-written, few suggestions need to revise below.
- Add comprehensive information about Garlic acid in the introduction
- AUTHOR provide In vivo activity of garlic acid in poultry
- Provide detail information/chemical composition of Garlic used in Poultry
- What are Bioactive Compounds and Biological Functions of Garlic, add detail
- Add a figure of the chemical structure of Garlic
- Add the mechanisms of the properties of garlic with different effects
- Provide list of abbreviation separately at the end of the review
Author Response
Dear respected reviewer
Thanks for your valuable comments that will add a great value and help to improve the article
1- Add comprehensive information about Garlic acid in the introduction
- The introduction contains general information on different components of garlic including garlic acid.
2- AUTHOR provide In vivo activity of garlic acid in poultry
- The different effects of garlic components including garlic acids in poultry have been provided in different sections of the article.
3- Provide detail information/chemical composition of Garlic used in Poultry
- Added in table (1) and figure (1).
4- What are Bioactive Compounds and Biological Functions of Garlic, add detail
- Added in the introduction section and figure 1.
5- Add a figure of the chemical structure of Garlic
- Added (figure 1).
6- Add the mechanisms of the properties of garlic with different effects
- The mechanisms of the properties of garlic with different effects have been included in each section of the article.
7- Provide list of abbreviation separately at the end of the review
- Added
Best regards
Round 2
Reviewer 1 Report
Comments and Suggestions for Authors
Nice job, overall.
Please make sure that figures are clear for the reader.
Author Response
Dear respected reviewer
Thanks for your kind revision and comments.
Please, find the attached revised article containing revision in green highlight.
Best regards
Prof. Dr. Wafaa Abd El-Ghany
